# Effect of Aqueous Extract of Maca Addition to an Extender for Chilled Canine Semen

**DOI:** 10.3390/ani12131638

**Published:** 2022-06-26

**Authors:** Chiara Del Prete, Alfonso Calabria, Valentina Longobardi, Veronica Palumbo, Barbara Merlo, Eleonora Iacono, Simona Tafuri, Domenico Carotenuto, Francesca Ciani, Sara Damiano, Roberto Ciarcia, Natascia Cocchia

**Affiliations:** 1Department of Veterinary Medicine and Animal Productions, University of Naples ‘Federico II’, Via Federico Delpino 1, 80137 Naples, Italy; alfonso.calabria@unina.it (A.C.); longobardivalentina@gmail.com (V.L.); veronica.palumbo@unina.it (V.P.); stafuri@unina.it (S.T.); ciani@unina.it (F.C.); sara.damiano@unina.it (S.D.); roberto.ciarcia@unina.it (R.C.); ncocchia@unina.it (N.C.); 2Department of Veterinary Medical Sciences, University of Bologna, Via Tolara di Sopra 50, Ozzano dell’Emilia, 40064 Bologna, Italy; barbara.merlo@unibo.it (B.M.); eleonora.iacono2@unibo.it (E.I.); 3Facultad de Ciencias Biologicas, Universidad Nacional Mayor San Marcos (UNMSM), Avenida Universitaria 34, Lima 15081, Peru; domenicar@alice.it

**Keywords:** canine spermatozoa, antioxidant, *Lepidium meyenii*, semen cooling, sperm hyperactivation

## Abstract

**Simple Summary:**

The wide use of artificial insemination in dogs justifies the development of new strategies to prevent the reduction of fertilizing ability of stored semen. In recent years, the use of plant antioxidant supplementation has become increasingly popular. Maca (*Lepidium meyenii*) is an Andean edible root with antioxidant properties. The effectiveness of the oral supplementation of Maca in improving fresh semen quality and quantity and cooling or freezing ability has already been reported. This is the first in vitro study on the effects of aqueous extract of Maca on canine spermatozoa. The addition of low concentrations of aqueous extract of Maca to the canine chilled extender had positive effects only until 24 h of storage, increasing hyperactivation of sperm cells and preserving DNA integrity of spermatozoa in short-term storage. Meanwhile, a high concentration of Maca had an immediately deleterious effect on semen quality.

**Abstract:**

Antioxidant supplementation has been proposed as a new strategy to improve the long-term preservation of semen. The aim of this study was to evaluate the effect of Maca supplementation of semen extender on quality-related canine semen parameters during cooling. Ejaculates from nine dogs were cooled for 7 days in the absence (control group) or in the presence of 10, 20 and 50 μL/mL of an aqueous extract of Maca. Sperm were evaluated for sperm viability, motility, DNA fragmentation and lipid peroxidation after 3 h, 24 h, 4 days and 7 days of storage. The addition of 10 μL/mL of Maca preserved sperm DNA and plasma membrane integrity at 3 h and increased sperm curvilinear velocity after 24 h. Treatment with 20 and 50 μL/mL of Maca increased the percentage of hyperactivated sperm after 3 h. Moreover, semen treated with 20 μL/mL of Maca decreased lipid peroxidation at 24 h. A significant reduction of sperm DNA and plasma membrane integrity as well as of kinetics parameters between 3 and 24 h of refrigerated storage with the higher concentration tested was observed. Although Maca was not able to protect canine semen with extended refrigeration storage time, it increased hyperactivation and preserved DNA integrity in short-term storage.

## 1. Introduction

The practice of artificial insemination with cooled-stored semen is widely utilized in dog breeding. The shipment of semen reduces the movement of animals and increases genetic variation, expanding the number of males available for breeding. The main issue with the use of chilled semen is that during storage at 4 °C, spermatozoa undergo changes that could affect their fertilizing ability [1]. This detrimental effect is related to oxidative stress due to an excess of reactive oxygen species (ROS) and a decrease in antioxidants [1,2]. Oxidative stress significantly damages sperm functions such as motility, fluidity of the sperm plasma membrane and the integrity of DNA due to lipid peroxidation induced by ROS [3,4]. Therefore, a feasible new strategy to improve the long-term preservation of semen is the use of antioxidants that keep only a small amount of ROS necessary to maintain normal sperm function [1,5,6]. The antioxidant supplementation has been proposed to reduce the impact of oxidative stress during the canine sperm storage process and slow or prevent semen deterioration [7,8,9,10,11].

In the last years, there has been a growing interest in natural antioxidants found in plants, and among the most popular supplements, Maca (*Lepidium meyenii* Walpers) has attracted global attention. Maca is an Andean edible root that grows exclusively between 3500 and 4500 m above sea level. Maca is classified into three ecotypes according to the color of the hypocotyls (red, yellow and black) that show different concentrations of metabolites and different biological activities [12]. Yellow Maca contains several bioactive secondary metabolites, such as glucosinolates and specific alkaloids, called Macamides, that are responsible for its antioxidant effect [13,14]. Several studies reported the effectiveness of the oral supplementation of Maca in improving fresh semen quality and quantity in humans, mice, bovines and stallions [13,15,16,17]. Moreover, semen of animals fed with Maca supplementation diets showed an improved cooling and freezing ability [17,18,19]. It has been recently demonstrated that Maca extract improves in vitro fertilization rates in mice by inducing an acrosome reaction and increasing sperm motility [20]. To the best of our knowledge, the supplementation of the semen extender for cooling or freezing with Maca has not yet been investigated.

Therefore, the aim of this study was to test for the first time the effect of different concentrations of aqueous extract of Maca on canine quality-related semen parameters (viability, motility, DNA fragmentation and oxidative stress) during storage at 4 °C for 7 days.

## 2. Materials and Methods

### 2.1. Animals

The semen samples included in this study were collected from dogs of the FOOF breeder center through their routine practice in the framework of breeding programs. The research involved nine dogs, 8 small breed (2 French bulldogs, 1 Jack Russel, 1 Pug, 1 Shih Tzu, 1 Poodle and 1 Cavalier King Charles Spaniel) and 1 large breed (Golden Retriever), with ages ranging from 1.5 to 8 years (median age was 6). Dogs received a standard commercial dog food twice daily and water ad libitum. All dogs received routine deworming treatments and vaccinations and shared the same environment for at least 6 months before the study.

The experiment was conducted in accordance with the code of ethics (D.lgs. 26—04/03/2014), and it was approved by the Ethics Committee of the Department of Veterinary Medicine and Animal Productions at the University of Naples Federico II, Italy (prot. no. PG/2021/0057934 of 07/06/2021).

### 2.2. Semen Collection and Processing

Semen collection was performed with an artificial vagina. Raw semen was evaluated for volume, color and concentration using a Burker’s counting chamber. Each ejaculate was split into 4 aliquots that were diluted to reach a final concentration of 100 × 10^s^ sperm/mL respectively in egg-yolk TRIS-citrate glucose (EYT-G: Tris 2.4 g, Citric Acid 1.4 g, Glucose 0.8 g, Penicillin G Sodium Salt 0.06 g, Streptomycin 0.1 g, 20 mL of egg yolk and distilled water to 100 mL), i.e., the control (CTRL group) and in YET-G supplemented with 10, 20, 50 μL/mL of maca extract (M10, M20 and M50 groups). All aliquots were placed in a syringe without air, transported to the laboratory at 4 °C within 3 h and then stored in the fridge at 4 °C for 7 days. Control and treated semen samples were evaluated for sperm viability, sperm motility, DNA fragmentation and lipid peroxidation after 3 h, 24 h, 4 days and 7 days (4 d and 7 d) of storage.

### 2.3. Maca Source and Preparation of Aqueous Extract of Maca

The yellow ecotype of Maca used in this study was acquired from the district of Junín, Andean highlands of Peru (4100 m above sea levels), with a taxonomic identification by Professor Carotenuto D. at the Universidad Nacional Mayor de San Marcos, Lima, Peru. Roots were treated according to the traditional open-field method of drying: exposition of the hypocotyls for two months at extreme temperature cycles, under intense light conditions, and atmospheric pressure typical of the high-altitude environment (>3500 m). After drying, hypocotyls were selected, washed and milled to flour with a particle size of 0.8 mm.

Aqueous extract of Maca was prepared in accordance with the method described by Fei et al. [21]. Fifty grams of Maca powder were mixed with 1000 mL of water and automatically stirred in a water bath at 70 °C for 3 h. After that, the solution was centrifugated at 4000 RPM for 10 min, and then, the extraction within a water bath at 70 °C for 2 h was repeated. The final solution was placed in small vials and stored in a refrigerator at 4 °C for further use. The final aqueous extract of Maca should have concentrations of 750 mg/mL.

A chemical analysis of the powder and the aqueous extract of Maca were performed through liquid Chromatography with tandem mass spectrometry (LC-MS-MS) at the Interuniversity Consortium Biostructures and Biosystems National Institute (INBB). Concentrations of different metabolites specific to Maca, such as Macaenes (polyunsaturated fatty acids), Macamides (a series of nonpolar and long-chain fatty acid N-benzylamides) and Macalines or Lepilidines were reported in Table 1.

### 2.4. Membrane Integrity (Hypo-Osmotic Swelling Test)

The hyposmotic swelling (HOS) test was carried out at each time point for the assessment of the functional integrity of the sperm plasma membrane in control and treated groups. Twenty microliters of semen were incubated at 37 °C for 45 min with 80 μL of pre-warmed HOS solution (0.73 g sodium citrate and 1.35 g fructose in 100 mL of distilled water, 150 mOsm). After incubation time, a volume of 10 μL was placed on a glass slide and covered with a cover slip. Evaluations were conducted under phase-contrast microscopy (40×; Eclipse E200, Nikon, Tokyo, Japan) by operators unaware of the experimental design. The cells were classified as positive (damaged membrane) or negative (intact membrane) according to the presence or absence of coiled tails, respectively. A total of 200 spermatozoa were counted.

### 2.5. Motility Assessment

Sperm motility parameters (total and progressive motility, sperm subpopulations and semen kinetic parameters) were assessed by Sperm Class Analyzer (SCA) system (Microptic SL, Veterinary Edition, Barcelona, Spain) installed on a camera-equipped light microscope system (Eclipse E200, Nikon, Tokyo, Japan). The following parameters were considered for the assessment: total motility (%), progressive motility (%), the percentage of sperm subpopulations (rapid and medium progressive), average path velocity (VAP; μm/s), straight-line velocity (VSL; μm/s), curvilinear velocity (VCL; μm/s), straightness (STR; %) and linearity (LIN; %).

SCA system settings for dog semen classified as spermatozoa all the particles sized between 10 and 80 μm^2^ and as progressively motile spermatozoa with 75% STR. The minimum velocity (VCL) values considered for slow-medium and rapid spermatozoa subpopulations were 50 and 100 μm/s; spermatozoa with VCL below 10 μm/s were considered static and spermatozoa with VCL > 150 μm/s and ALH >3.5 μm as hyperactive. Sixty frames per second with minimum contrast of 35 were acquired.

For the evaluation, an aliquot of control or treated (M10, M20 and M50) semen at each time point (3 h, 24 h, 4 d and 7 d) was diluted 1:3 with TRIS-glucose-citrate in order to reach a concentration of 30 × 10^6^ sperm/mL as required by SCA system and incubated at 37 °C for 10 min before evaluation. Then, 5 µL were spotted onto a pre-warmed glass microscope slide, covered with a glass coverslip (22 mm × 22 mm). At least 500 sperm cells in five randomly selected fields were evaluated.

### 2.6. DNA Fragmentation

Sperm DNA fragmentation was examined in each sample by using the terminal deoxynucleotidyl transferase dUTP nick end labeling (TUNEL) assay or in-situ Cell Death Detection Kit (Sigma-Aldrich, St. Louis, MO, USA), as previously described by Longobardi et al. [22]. Briefly, 40 μL of cooled semen was fixed with 250 μL of 4% (*w*/*v*) paraformaldehyde in phosphate-buffered saline (PBS; pH 7.4) for 45 min at room temperature. After incubation, sperm cells were washed twice (300 g × 15 min) with PBS with Polyvinylpyrrolidone (PVP;1 mg/mL). After supernatant aspiration, the pellet was diluted at 1:10 with PBS. A drop of semen (approximately 20 μL) was smeared on an object glass, dried, and permeabilized with 0.1% Triton X-100 in 0.1% sodium citrate for 10 min. Then, slides were washed twice with PBS, air-dried, and incubated with a TUNEL reaction mixture for 1 h at 37 °C in a humidified atmosphere in the dark. The negative control was made by adding all components of the label solution (except for the terminal deoxynucleotidyl transferase enzyme), and the positive control was made by incubating the samples with DNAse recombinant to induct DNA separation for 10 min before incubation with the Tunel reagent. After one h of incubation, slides were stained with PBS -PVP labeled with 1 mg/mL Hoechst 33342, for 30 min, at room temperature and then rinsed with PBS. The results were examined using a fluorescent microscope (Eclipse E-600; Nikon, Tokyo, Japan) under ultraviolet light; the excitation wavelength was 460 nm for the blue fluorescence and 520 nm for the green fluorescence.

TUNEL assay evaluates the presence of free 3′-hydroxyl ends, which are identified by terminal deoxynucleotidyl transferase (TdT) enzyme and catalyze the addition of fluorescently labeled deoxyuracil triphosphate breaks in DNA strands. Spermatozoa in blue (Hoechst+) in a bright green fluorescence (Tunel+) showed damaged (fragmented) DNA, while spermatozoa in a dull green fluorescence showed normal DNA.

### 2.7. Lipid Peroxidation

Sperm lipid peroxidation of control and treated samples at each time point was determined by assaying the Malondialdheyde (MDA) concentration by means of the thiobarbituric acid (TBA) test [23]. In order to precipitate proteins, 100 μL of each sample was treated with 0.5 mL of cold 30% (*w*/*v*) trichloroacetic acid and centrifugate. One millimeter of supernatant was reacted with 1.3 mL of 0.5% (*w*/*v*) TBA at 85 °C for 40 min. In the TBA test reaction, each molecule of MDA reacts with two molecules of TBA with the production of a pink pigment having maximal absorbance at 532–535 nm. After cooling, the fluorescence was read at wavelengths of 536 nm for excitation and 557 nm for emission using a SPEX Fluoromax spectrophotofluorimeter (GloMax^®^-Multi Detection System, Promega, Madison, WI, USA). Concentrations of MDA calculated using a calibration curve ranged between 0.5–2 pmoles/mL and were expressed as nmol/L of proteins.

### 2.8. Statistical Analysis

Data were first recorded using a computerized spreadsheet (Microsoft^®^ Excel^®^ 2021, Redmond, WA, USA) and then imported into Statistical Package for Social Sciences (SPSS IBM^®^ Statistics version 27.0, IBM Corporation, Armonk, NY, USA) for statistical analysis. The Kolmogorov–Smirnov test was utilized for normality analysis of the parameters. All data were expressed in median and interquartile ranges (IQR) for the violation of normality. Non-parametric tests were used for evaluation. The effect of storage time on sperm analysis data was evaluated in each group (control, M10, M20 and M50) using a Friedman test, and in case of significance, post hoc analysis with Wilcoxon’s signed-rank test was conducted to compare individual storage times; differences between groups at each time point were also evaluated with Wilcoxon’s signed-rank test. Differences were considered statistically significant when *p* ≤ 0.05.

## 3. Results

### 3.1. Fresh Semen

All ejaculates collected were white and milky in consistency. The median (IQR) of the volume of the sperm-rich fraction was 2.5 (1.5–5.0) mL with a sperm concentration of 230 (174–396) × 10^6^ sperm/mL.

### 3.2. Membrane Integrity (HOS)

With regard to storage time, a decrease (*p* ≤ 0.05) of membrane integrity was only observed at 7 days in the M50 group. No differences in membrane integrity were found between the treated groups and the control at 3 h, 4 d and 7 d. In M50-treated group, a decrease (76.5 (76–85)%; *p* ≤ 0.05) was recorded compared to M10 (80.5 (80–88)%) and M20-treated groups (80 (77–84.5)%) at 3 h. Likewise, at 24 h a reduced sperm integrity was found in M50 (76.5 (72–83.7)%; *p* ≤ 0.05) compared to CTRL group (83 (77–84.5)%) and M20-treated group (77 (75.6–86.2)%).

### 3.3. Motility

Figure 1 represents the results of total and progressive motility during storage time and the differences between time points inside each group and among groups at each time point. Semen motility gradually reduced during preservation at 4 °C in the control, as well as in the Maca-treated groups, as shown in Figure 1. After 4 days of cooling, total motility was significantly lower in the M10-treated group than in the CTRL group and in the M50- than in the M20-treated group (*p* ≤ 0.05). Moreover, at the same time point, progressive motility was lower in the M50-treated group compared with the CTRL group (*p* ≤ 0.05).

Analysis of sperm subpopulations is reported in Table 2. After 24 h, semen stored at 4 °C with the addition of 10 μL/mL of Maca had an increase in rapid sperm cells, although not significant. The percentage of rapid sperm cells remained constant until 4 days of storage at 4 °C in the CTRL group; meanwhile, at 4 days, there was a decrease in treated groups (*p* ≤ 0.05). In all groups, the proportion of rapid sperm cells declined at 7 days (*p* ≤ 0.05). On the other hand, rapid progressive sperm were lower in the M50-treated group than in all the other CTRL and treated groups at 4 days of sperm storage (*p* ≤ 0.05). The percentage of medium progressive spermatozoa decreased after 4 d in all groups.

Sperm hyperactivation during cooling preservation was different in all groups, as illustrated in Table 2. Treatment with 20 and 50 μL/mL of Maca increased the percentage of hyperactivate sperm compared to CTRL after 3 h of storage (*p* ≤ 0.05). On the contrary, after 4 days of storage at 4 °C, the percentage of hyperactivated sperm decreased in the M50-treated group compared to the CTRL group (*p* ≤ 0.05).

As shown in Table 3, the temporal decrease in kinetic parameters showed a similar trend in the different groups, except for VCL. This parameter increased after 24 h of storage (*p* ≤ 0.05), only in M10-treated group. With regard to kinetic parameters, after 3 h at 4 °C, the CTRL group had higher VSL, VAP, LIN and STR than the M50-treated group (*p* ≤ 0.05) and higher STR than the M20-treated group (*p* ≤ 0.05). Moreover, after 24 h of cooling, semen treated with 50 μL/mL of Maca (M50 group) had reduced (*p* ≤ 0.05) VCL than semen diluted with the lowest concentration of Maca (M10 group)

### 3.4. DNA Fragmentation

Results of DNA fragmentation are reported as median (IQR) and range (min–max) in Figure 2A. For all groups, storage time has an effect on DNA fragmentation (*p* ≤ 0.05), especially after 7 days of cooling. Differences in DNA fragmentation between groups were found at 3 h and at 7 days of refrigeration. After 3 h of cooling, sperm DNA fragmentation was lower in the M10-treated group than in CTRL (*p* ≤ 0.05) and M50 groups (*p* ≤ 0.05). After 7 days of refrigeration of sperm, DNA fragmentation increased (*p* ≤ 0.05) in M50-treated group compared to the other two treated groups (M10 and M20).

### 3.5. Lipid Peroxidation

As displayed in Figure 2B, MDA concentrations increased in all treated groups after 7 d (*p* ≤ 0.05), However, after 24 h of cooling, lipid peroxidation was lower in all Maca-treated groups (M10, M20 and M50) with respect to CTRL, although the difference was only significant between M20-treated group and CTRL group (*p* ≤ 0.05).

## 4. Discussion

This study shows for the first time the in vitro effects of Maca on canine spermatozoa preserved for 7 days at 4 °C. The cytoprotective effects of Maca have been demonstrated to be associated with its antioxidant activity by increasing the enzyme activity of superoxide dismutase and by direct free radical scavenging [24,25]. As expected, during refrigeration in both control and Maca-treated groups, there was a progressive and significant reduction in semen quality. One of the factors that cause this impairment is the imbalance between oxidants and antioxidants [26]. The protective action of the addition of antioxidants in canine semen extender for chilling has been already tested, suggesting that Lycopene or Vitamin E and B16 are able to preserve semen quality of chilled dog spermatozoa [10,27]. Antioxidants counteract oxidative stress, improving membrane integrity and motility and preventing lipid peroxidation and DNA fragmentation of spermatozoa [27,28,29]. The results of this study indicate that Maca had a protective effect on canine chilled semen until 24 h of storage. Particularly, the semen extender treatment with 10 μL/mL preserved DNA and plasma membrane integrity of spermatozoa at 3 h of storage and after 24 h of storage, significantly increased VCL of spermatozoa and improved the percentage of rapidly progressive and hyperactivated sperm cells, albeit not in a significant manner. A previous study that investigated the in vitro effect of Maca on human spermatozoa reported an increase in total motility but not in VCL or in other kinetic parameters [20]. The color of Maca hypocotyls or the use of different methods of cultivation, processing and extraction (methanol, chloroform, DMSO and water) of Maca can change concentrations of Maca bioactive metabolites [13,30,31]. The content of secondary metabolites (Macamides, Macaenes and Lepilidines) improves the cold shock resistance of spermatozoa [15,17,32]. For this reason, we decided to investigate the quantities of some constituents of the aqueous extract of Maca used in this study and to test the supplementation of the semen extender with three different concentrations.

Spermatozoa are highly prone to peroxidative damage due to the higher polyunsaturated fatty acid contents. MDA concentration is indicative of lipid peroxidation as a marker of oxidative stress, and it is an accepted diagnostic tool for humane infertility workup [33,34,35]. Previous studies reported the high susceptibility of canine epididymal sperm to the deleterious effect of hydrogen peroxide, MDA and hydroxyl radical [36]. The MDA can exert a damaging effect by combining with other molecules such as proteins, DNA and RNA and provoking a reduction in sperm viability, motility and DNA integrity [37,38,39,40]. This study observed that an increase in the main byproduct of lipid peroxidation after 7 days of cooling determined a meaningful reduction of total and progressive motility and all kinetic parameters. Instead, DNA fragmentation and membrane integrity were not correlated with high MDA concentrations. Moreover, the semen with the semen extender addition of Maca had lower levels of MDA at 3 and 24 h of cooling storage, the only statistical difference compared at 24 h between control and semen extender with the addition of 20 μL/mL. From these results, we can only speculate that the addition of Maca had a protective role against oxidation, preventing the formation of lipid peroxidation. More cases and further studies on the effect of Maca during canine semen freezing are needed to clarify whether Maca could prevent the cold shock of spermatozoa.

In this study, contrasting and opposite effects at different Maca concentrations demonstrated dose-related effects of Maca. High concentrations of Maca affected sperm parameters by increasing DNA fragmentation and damaged membranes and also reducing some kinetic parameters (VCL, VSL, VAP, STR and LIN) between 3 and 24 h of storage at refrigerating temperatures. It was reported that Macamides and Macaenes have cytotoxicity against different cancer cell lines by inducing apoptosis [31,41,42,43]. Cells undergoing apoptosis usually show several cellular changes, including DNA fragmentation and membrane disruption. Probably the toxicity of Maca on spermatozoa is expressed not only with increased apoptosis but also by reducing sperm velocities.

The addition of all concentrations of Maca to semen extender increased the hyperactivation of sperm cells at 3 h, but significance was apparent only in the semen with the addition of 20 and 50 μL/mL of Maca. A previous study reported that the interaction of human and mouse sperm with Maca tends to increase the amplitude of lateral head displacement, which is strictly correlated with hyperactivation [20,44]. Alkaloid components of Maca could be responsible for the hyperactivation of the spermatozoa, as already described after the addition of caffeine alkaloids to the semen [45,46]. It has been described that the alkaloids contained in caffeine that are similar to those of Maca provoke sperm hyperactivation by increasing intracellular cyclic adenosine monophosphate (cAMP) or by promoting activation of calcium ion-permeable cation channels in the plasma membrane of sperm [45,46,47]. Further analysis should be conducted to clarify the responsibilities and the mechanism underlying this effect of Maca. When hyperactivated, spermatozoa are able to swim through viscoelastic substances and successfully penetrate the zona pellucida; indeed, it is associated with increased sperm fertilizing capacity [48,49]. For these reasons the hyperactivation is required when sperm enter the uterus and not during storage because this event determines an overall energy consumption reducing sperm lifespan.

## 5. Conclusions

The main findings emerging from the results of this study are that the addition of 10 or 20 μL/mL of aqueous extract of Maca to the chilled extender had positive effects until 24 h of storage, while the highest concentration of Maca tested in this study (50 μL/mL) had an immediately deleterious effect on quality-related semen parameters. Maca cannot sufficiently protect canine semen for extending refrigeration storage time; however, low concentrations of Maca could be proposed before insemination in fresh, cooled or freeze-thawed semen to increase hyperactivation of sperm cells and to preserve DNA integrity of spermatozoa until fertilization.

## Figures and Tables

**Figure 1 animals-12-01638-f001:**
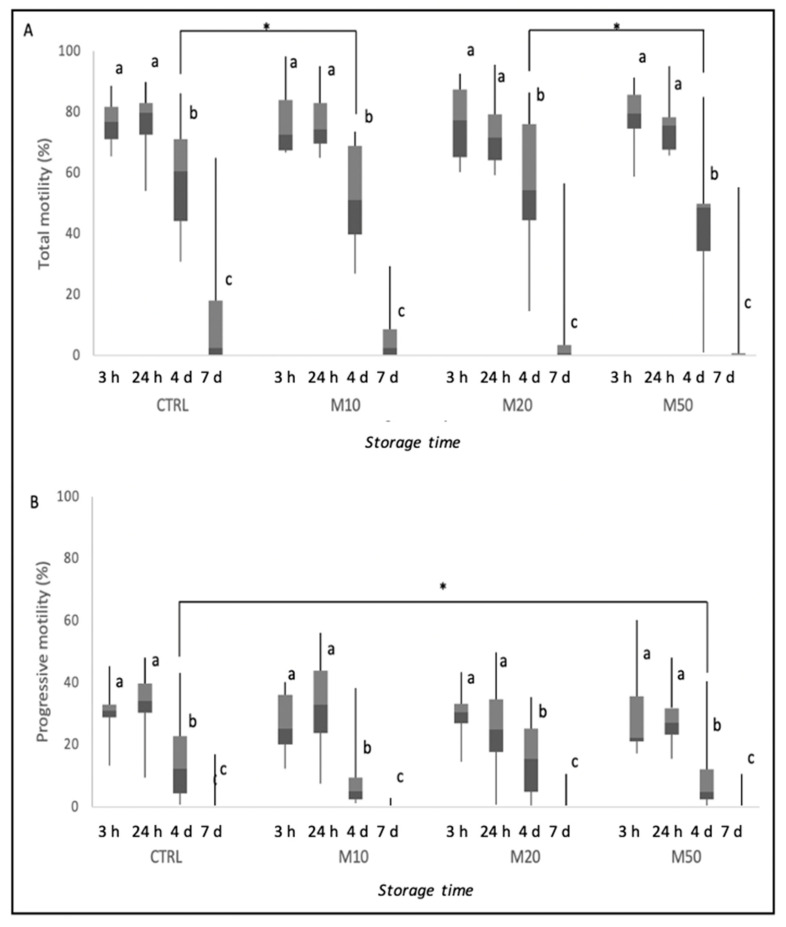
Total (**A**) and progressive motility (**B**) of dog semen (*n* = 9) diluted with only egg-yolk tris-citrate glucose (YET-G; CTRL) or with EYT-G with the addition of three different concentrations (10, 20 and 50 μL/mL) of aqueous extract of Maca (M10, M20 and M50) stored at refrigeration temperature (4 °C) and evaluated at 3 h, 24 h, 4 d and 7 d of storage. Asterisk above the bar indicates statistical difference between groups at *p* ≤ 0.05. The letters a, b and c indicate statistically significant differences at *p* ≤ 0.05 between time points within each group.

**Figure 2 animals-12-01638-f002:**
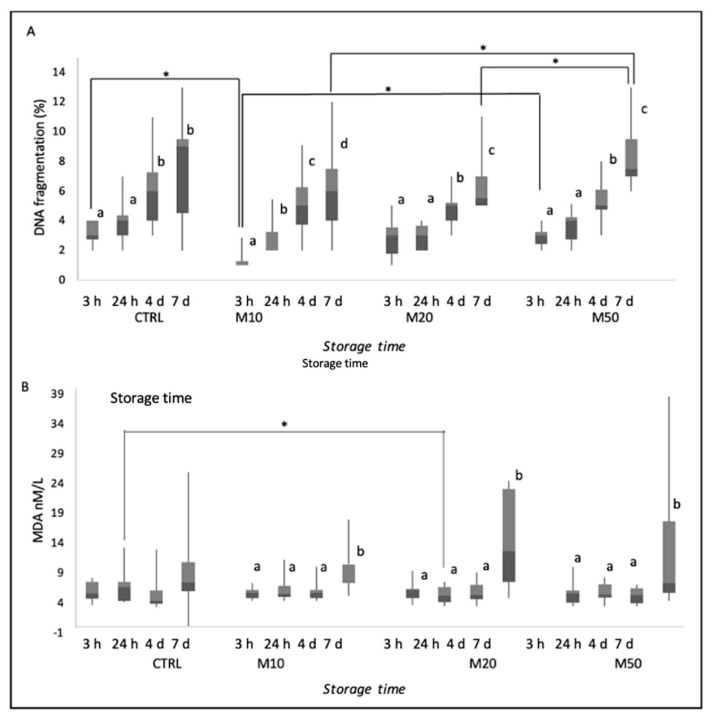
Percentage of DNA fragmentation (**A**) and lipid peroxidation (**B**) of dog semen (*n* = 9) diluted with only egg−yolk tris−citrate glucose (EYT−G; CTRL) or with EYT−G with the addition of three different concentrations (10, 20 and 50 μL/mL) of aqueous extract of Maca (M10, M20 and M50) stored at refrigeration temperature (4 °C) and evaluated at 3 h, 24 h, 4 d and 7 d of storage. Asterisk above the bar indicates statistical difference between groups at *p* ≤ 0.05. The letters a, b, c and d indicate statistically significant differences at *p* ≤ 0.05 between time points within each group.

**Table 1 animals-12-01638-t001:** Concentrations of Maca metabolites in Maca powder and aqueous extract of Maca.

Chemical Structures of Maca Metabolites	Maca Powder (μg/L)	Aqueous Extract of Maca (μg/L)
5-oxo-6E,8E-octadecadienoic acid (Macaen)	69.53	17.89
N-(3-hydroxy-benzyl)-2Z-fivecarbon acrylamide	614.29	157.99
N-benzyl-5-oxo-6E,8E-octadecadienamide (MI 7)	46.08	61.81
N-benzyloctadecanamide (MI 16)	53.96	28.89
1,3-dibenzyl-2, pentyl-4, 5-trimethylimidazilium (Lepilidine A)	59.03	13.31
(1R,3S)-1-methyltetrahydro-beta-5,6-hydridecarboline-3-carboxylic acid (MTACA)	47.17	3.63
1-dibenzyl-2-propane-4,5-dimethylimidazilium	19.52	1.25

**Table 2 animals-12-01638-t002:** Sperm subpopulations (% of rapid, medium and hyperactivated spermatozoa) of canine (*n* = 9) semen of control (CTRL) and semen extender Maca treatment groups (M10, M20 and M50) during storage at 4 °C for 7 days. All values are expressed as median and interquartile range (IQR).

	Storage Time			
Rapid progressive (%)	3 h	24 h	4 d	7 d
CTRL	14.4 (4.5–20.7) ^a^	17.8 (8.1–29.7) ^a^	6.5 (1.9–16) ^a,x^	0 (0–1.3) ^b^
M10	10.6 (7.4–17.6) ^a^	24.2 (6.6–29.1) ^a^	2.1 (1–4) ^b,xy^	0 (0–0) ^c^
M20	12.2 (5.6–21.8) ^a^	11.2 (5.3–20.2) ^a^	5 (1.5–11) ^b,x^	0 (0–1.5) ^c^
M50	13.3 (9.4–20.8) ^a^	12.6 (6–20.1) ^a^	1.8 (0.5–2.4) ^b,y^	0 (0–0) ^c^
**Medium progressive (%)**				
CTRL	14.3 (13.4–21.7) ^a,xy^	12.3 (7.3–23.2) ^a^	2.4 (1.7–13.8) ^b^	0 (0–1.5) ^c^
M10	13.4 (6–20.7) ^a,xy^	12 (6.2–21.5) ^a^	2.1 (1–7.5) ^b^	0 (0–0.1) ^c^
M20	12 (11.9–21.1) ^a,x^	15.2 (8.1–21) ^a^	10.5 (0.9–15.5) ^b^	0 (0–0.5) ^c^
M50	14.5 (9–20.1) ^a,y^	13 (7.7–22.4) ^a^	2.5 (0.3–12.9) ^b^	0 (0–0) ^c^
**Hyperactive sperm (%)**				
CTRL	1.6 (0.9–3.7) ^a,x^	1.3 (0.9–4.6) ^a^	1.1 (0.6–4.4) ^a,x^	0 (0–0.14) ^b^
M10	3.5 (2.2–8.1) ^a,xy^	2.88 (1.8–4.1) ^a^	0.5 (0–1.2) ^b,xy^	0 (0–0) ^c^
M20	3.1 (1.4–5.9) ^a,y^	2.2 (0.7–3.8) ^a^	0.25 (0–1.6) ^b,xy^	0 (0–0.2) ^b^
M50	2.1 (1.1–6.7) ^a,y^	1.7 (1.2–4) ^a^	0 (0–0.8) ^b,y^	0 (0–0) ^b^

The letters a, b and c indicate statistically significant differences at *p* ≤ 0.05 between time points within each row (group; letters x and y represent statistical differences among groups within each column (time point).

**Table 3 animals-12-01638-t003:** Trends of semen kinetic parameters of canine (*n* = 9) semen of control (CTRL) and semen extender Maca treatment groups (M10, M20 and M50) during storage at 4 °C for 7 days. All values are expressed as median and interquartile range (IQR).

	Storage Time			
VCL	3 h	24 h	4 d	7 d
CTRL	85.4 (82.2–86.3) ^a^	86.1 (84.6–86.6) ^a,xy^	86.7 (74.8–88.2) ^a^	0 (0–76.8) ^b^
M10	85.7 (81.3–87.8) ^a^	87.8 (86.1–89.4) ^b,x^	82.4 (78.2–84) ^a^	0 (0–33.8) ^c^
M20	84.2 (82.2–87.7) ^a^	86.2 (82–87.6) ^a,xy^	84.7 (40.5–85.8) ^a^	0 (0–41) ^b^
M50	85 (81.7–87.1) ^a^	83.8 (80.2–86.2) ^a,y^	78.9 (38.3–84.1) ^a^	0 (0–0) ^b^
**VSL**				
CTRL	61 (56.3–66.4) ^a,x^	55.4 (48.5–62.2) ^a^	44.7 (35.7–50.3) ^b^	0 (0–27.5) ^c^
M10	60.6 (50–64.5) ^a,xy^	56 (49.7–63.6) ^a^	40.1 (34.2–46) ^b^	0 (0–19.3) ^c^
M20	59 (55.4–61.7) ^a,xy^	57.3 (46.6–61.8) ^a^	42.9 (19.5–51.5) ^b^	0 (0–21.7) ^c^
M50	57.9 (50.8–64.4) ^a,y^	49.9 (44.5–59.7) ^a^	43.2 (16.3–51.3) ^b^	0 (0–0) ^c^
**VAP**				
CTRL	66.4 (61.8–70.7) ^a,x^	61.2 (55.2–67.1) ^a^	51.5 (42.8–57) ^b^	0 (0–34.4) ^c^
M10	66.5 (56.2–69.6) ^a,xy^	62.3 (57.2–68.7) ^a^	47.6 (42–52.9) ^b^	0 (0–22.6) ^c^
M20	65 (63–66.6) ^a,xy^	63.4 (53.4–68.1) ^a^	50.5 (23.6–57.6) ^b^	0 (0–24) ^c^
M50	64.1 (57.2–69.5) ^a,y^	58.8 (51.7–64.8) ^a^	50.6 (19.8–57.9) ^b^	0 (0–0) ^c^
**LIN**				
CTRL	71.6 (66.2–79.1) ^a,x^	67.4 (56.9–72.1) ^a^	51.6 (40.3–62.4) ^b^	0 (0–35.4) ^c^
M10	73 (59.8–75.5) ^a,xy^	65.4 (55.5–72.2) ^a^	48.9 (41.5–54.1) ^b^	0 (0–25) ^c^
M20	70.1 (65.3–73.6) ^a,xy^	66.3 (57.3–72.5) ^a^	50.9 (2.9–60.1) ^b^	0 (0–24.2) ^c^
M50	66.7 (61.2–73.8) ^a,y^	60.2 (55–71.9) ^a^	50.3 (20.9- 63.6) ^b^	0 (0–0) ^c^
**STR**				
CTRL	91.4 (90.4–93.1) ^a,x^	90 (87.5–92) ^b^	86.3 (82.5–87.8) ^c^	0 (0–80.4) ^d^
M10	90.9 (88–92.3) ^a,xy^	89.7 (85.6–92.1) ^a^	83.9 (80.6–86.5) ^b^	0 (0–41.9) ^c^
M20	90.6 (87.5–92.4) ^a,y^	89.9 (88.8–92.8) ^a^	84.9 (40.5–88.2) ^b^	0 (0–41.8) ^c^
M50	89.7 (87.4–92.2) ^a,y^	86.3 (85–91.4) ^b^	85.2 (41–92.2) ^b^	0 (0–0) ^c^

The letters a, b, c and d indicate statistically significant differences at *p* ≤ 0.05 between time points within each row (group); letters x and y represent statistical differences among groups within each column (time point).

## Data Availability

Not applicable.

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
