# Peer review of "Effect of Aqueous Extract of Maca Addition to an Extender for Chilled Canine Semen"

_animals, 2022, doi:10.3390/ani12131638_

Round 1
Reviewer 1 Report
line 144, 106 sperm/ml
in the text of each table should clarify what the numbers in the brackets are?
you could clarify what the aterisks means in the figures and you should present more clrearly the results
Author Response
Dear Reviewers,
We wish to thank you all for the time spent in reviewing our manuscript, and for all the priceless advice you gave us. We tried to critically answer to all your indications, and we feel that the manuscript has been highly improved. We hope it will be now worth for publication on Animals.
Please find below all the answer to Reviewers comments. All changes have been marked up using the “Track Changes” function of Word. All Authors have reviewed and approved the manuscript in this form.
Thank you in advance for your kind assistance,
Kind Regards
Dr. Chiara Del Prete
DVM, PhD. ECAR (biotechnology) Resident
Università di Napoli Federico II
Naples, Italy
Reviewer: 1
line 144, 106 sperm/ml
- Thank you, the change has been made.
in the text of each table should clarify what the numbers in the brackets are?
- Thank you for pointing this out. We add in the caption “All values are expressed as median and interquartile range (IQR).”
you could clarify what the aterisks means in the figures and you should present more clrearly the results
- Thank you for your suggestions, we add in the caption of the figures “Asterisk above the bar indicates statistical difference between groups at P ≤0.05”. We also try to make the results clearer, we hope that meet the requirements.
Reviewer 2 Report
General comments
The manuscript brings a new approach to improving the quality of dog semen preserved in liquid state, by addition of Maca extract (a plant derived antioxidant) to the semen extender. The introduction provides the background for the work, the methods are appropriate and well designed and the article is generally well written. As we can see in the Results section, Maca has a positive short-term (up to 24 h) effect on some sperm quality parameters and is unlikely to be used for longer storage of chilled semen. Maca supplementation increases the percentage of hyperactivated sperm, which is not beneficial during sperm storage, however it is associated with sperm cells capacitation and crucial for fertilization process. In my opinion the ability of Maca to increase sperm hyperactivation is important, promising and may find application in preparing sperm cells both for in vivo and in vitro fertilization procedure.
Detailed comments
Abstract:
line 27: there is no need to use the "sperm" twice in the same sentence, it is better to write: Sperm were evaluated for viability, motility …
lines 32-33: the sentence: A significant reduction of sperm DNA ….… higher concentration tested” - is not clear, it would be clearer if it ended with a passive verb (e.g. was observed/found)
Results
3.2. Membrane integrity - the results of membrane integrity could be presented in the table
line 213 – Figure X ? - did you mean Figure 1 ?
Figure 1 - statistical marks should be explained in the figure legend
line 254 - (P0.05) ? - did you mean P<0.05 ?
Figure 2 - statistical marks should be explained in the figure legend
Author Response
Dear Reviewers,
We wish to thank you all for the time spent in reviewing our manuscript, and for all the priceless advice you gave us. We tried to critically answer to all your indications, and we feel that the manuscript has been highly improved. We hope it will be now worth for publication on Animals.
Please find below all the answer to Reviewers comments. All changes have been marked up using the “Track Changes” function of Word. All Authors have reviewed and approved the manuscript in this form.
Thank you in advance for your kind assistance,
Kind Regards
Dr. Chiara Del Prete
DVM, PhD. ECAR (biotechnology) Resident
Università di Napoli Federico II
Naples, Italy
REVIEWER 2:
The manuscript brings a new approach to improving the quality of dog semen preserved in liquid state, by addition of Maca extract (a plant derived antioxidant) to the semen extender. The introduction provides the background for the work, the methods are appropriate and well designed and the article is generally well written. As we can see in the Results section, Maca has a positive short-term (up to 24 h) effect on some sperm quality parameters and is unlikely to be used for longer storage of chilled semen. Maca supplementation increases the percentage of hyperactivated sperm, which is not beneficial during sperm storage, however it is associated with sperm cells capacitation and crucial for fertilization process. In my opinion the ability of Maca to increase sperm hyperactivation is important, promising and may find application in preparing sperm cells both for in vivo and in vitro fertilization procedure.
Detailed comments
Abstract:
line 27: there is no need to use the "sperm" twice in the same sentence, it is better to write: Sperm were evaluated for viability, motility …
lines 32-33: the sentence: A significant reduction of sperm DNA ….… higher concentration tested” - is not clear, it would be clearer if it ended with a passive verb (e.g. was observed/found)
Results
3.2. Membrane integrity - the results of membrane integrity could be presented in the table
line 213 – Figure X ? - did you mean Figure 1 ?
Figure 1 - statistical marks should be explained in the figure legend
line 254 - (P0.05) ? - did you mean P<0.05 ?
Figure 2 - statistical marks should be explained in the figure legend
- Thank you for all your suggestions, all changes have been made, moreover we add in the caption of the figures the phrase“Asterisk above the bar indicates statistical difference between groups at P ≤0.05”.
Reviewer 3 Report
The authors tested the use of Maca in semen extender on the quality of canine spermatozoa to enhance viability and subsequent fertility of the cells upon warming/thaw for artificial insemination. Preservation of sperm cells has a broad application among many different species; therefore, identifying additives or supplements that have cooling or cryoprotective effects may prove to be very valuable information. The manuscript does require minor edits and additions to clarify the content of the study.
1. No title for Table 1. Left the manuscript template for the title.
2. There was no mention of adding semen extender to the ejaculate in the materials and methods; however, it was referred to frequently in the discussion. Please clarify that an extender was used and all of the components of the semen extender.
3. Line 118: Hyposmotic is capitalized, should be lower case.
4. Figure 1: superscripts and asterisk need to be defined in the figure title. Egg yolk concentrations need to be separated by commas or the word ‘and’ to be consistent with other figures. Shading of the mean bars is too dark and difficult to see the difference within bars.
5. Figure 2: Title is incomplete fragment; a “T’ is before the word “Percentage”. Was it supposed to be “The percentage”? The X axis titles are not centered. Superscripts and asterisk need to be defined in the figure title
6. P values: in throughout text are inconsistent in the spacing relative to the mathematical symbol, P, and numbers OR are missing the mathematical symbol.
7. Subunit fonts are inconsistent. For example: Table 1 has ug/L and the text has ml and mm/s. Micro subunits should be a mu not an actual u.
8. Table 2: reference to groups is unclear. There are different treatments and different times. Clarify that the ‘groups’ are referring to semen extender treatment groups in the table. It would be helpful to use the term ‘treatments’ in the text in the results section. Also needs to be on the same page.
9. Table 3: The table state that the parameters are ‘kinetic parameters’; however, in the materials and methods they are defined and described as sperm motility parameters. The table should state ‘motility parameters’ rather than kinetic OR change the text in materials and methods to accurately reflect the table.
10. Line 314: Sentence makes no sense. “effects emerged’ rather than ‘effects were emerged’…’relative to maca concentrations’ rather than ‘if were added low or high concentration of maca’. Please revise for clarification.
Author Response
Dear Reviewers,
We wish to thank you all for the time spent in reviewing our manuscript, and for all the priceless advice you gave us. We tried to critically answer to all your indications, and we feel that the manuscript has been highly improved. We hope it will be now worth for publication on Animals.
Please find below all the answer to Reviewers comments. All changes have been marked up using the “Track Changes” function of Word. All Authors have reviewed and approved the manuscript in this form.
Thank you in advance for your kind assistance,
Kind Regards
Dr. Chiara Del Prete
DVM, PhD. ECAR (biotechnology) Resident
Università di Napoli Federico II
Naples, Italy
REVIEWER 3:
The authors tested the use of Maca in semen extender on the quality of canine spermatozoa to enhance viability and subsequent fertility of the cells upon warming/thaw for artificial insemination. Preservation of sperm cells has a broad application among many different species; therefore, identifying additives or supplements that have cooling or cryoprotective effects may prove to be very valuable information. The manuscript does require minor edits and additions to clarify the content of the study.
1.No title for Table 1. Left the manuscript template for the title.
- Thank you for pointing this out. We add a correct caption to the Table.
- There was no mention of adding semen extender to the ejaculate in the materials and methods; however, it was referred to frequently in the discussion. Please clarify that an extender was used and all of the components of the semen extender.
- Thank you for pointing this out. The semen extender used in this study is an egg-yolk TRIS-citrate glucose, as mentioned at line 91, we added the recipe for the preparation as suggested.
- Line 118: Hyposmotic is capitalized, should be lower case.
- Thank you, the change has been made.
- Figure 1: superscripts and asterisk need to be defined in the figure title. Egg yolk concentrations need to be separated by commas or the word ‘and’ to be consistent with other figures. Shading of the mean bars is too dark and difficult to see the difference within bars.
- Thank you for all your suggestions, all changes have been made, moreover we add in the figure title the phrase “Asterisk above the bar indicates statistical difference between groups at P ≤0.05. The letters a, b and c indicate statistically significant difference at P≤ 0.05 between time points within each group.”
- Figure 2: Title is incomplete fragment; a “T’ is before the word “Percentage”. Was it supposed to be “The percentage”? The X axis titles are not centered. Superscripts and asterisk need to be defined in the figure title.
- Thank you for pointing this out, we delete the “T” that was a typo and corrected the X axis title. Moreover, we add in the figure title the phrase “Asterisk above the bar indicates statistical difference between groups at P ≤0.05. The letters a, b and c indicate statistically significant difference at P≤ 0.05 between time points within each group.”
- P values: in throughout text are inconsistent in the spacing relative to the mathematical symbol, P, and numbers OR are missing the mathematical symbol.
- Thank you for pointing this out, we correct P values throughout the text.
- Subunit fonts are inconsistent. For example: Table 1 has ug/L and the text has ml and mm/s. Micro subunits should be a mu not an actual u.
- Thank you, the changes have been made.
- Table 2: reference to groups is unclear. There are different treatments and different times. Clarify that the ‘groups’ are referring to semen extender treatment groups in the table. It would be helpful to use the term ‘treatments’ in the text in the results section. Also needs to be on the same page.
- Thank you for pointing this out, we changed in “semen extender Maca treatment groups”.
- Table 3: The table state that the parameters are ‘kinetic parameters’; however, in the materials and methods they are defined and described as sperm motility parameters. The table should state ‘motility parameters’ rather than kinetic OR change the text in materials and methods to accurately reflect the table.
-Thank you for you suggestions, however we use the term “kinetic parameters” in case of velocities and sperm motility parameters when we talk about all the semen evaluation by SCA system. We add in the text at line 153: “ Sperm motility parameters (total and progressive motility, sperm subpopulations and semen kinetic parameters)…”.
- Line 314: Sentence makes no sense. “effects emerged’ rather than ‘effects were emerged’…’relative to maca concentrations’ rather than ‘if were added low or high concentration of maca’. Please revise for clarification.
- Thank you for pointing this out, we change with “ In this study, contrasting and opposite effects at different Maca concentrations, demonstrating dose related effect of Maca.”